# COVID-19 Vaccination in Patients with Myasthenia Gravis: A Single-Center Case Series

**DOI:** 10.3390/vaccines9101112

**Published:** 2021-09-29

**Authors:** Zhe Ruan, Yonglan Tang, Chunhong Li, Chao Sun, Ying Zhu, Zhuyi Li, Ting Chang

**Affiliations:** Department of Neurology, Tangdu Hospital, The Fourth Military Medical University, Xi’an 710038, China; ruanzhe573291596@126.com (Z.R.); tangyl2011@163.com (Y.T.); lichunhong0581@163.com (C.L.); biyfhvk666@163.com (C.S.); zz15399088127@163.com (Y.Z.)

**Keywords:** myasthenia gravis, COVID-19, vaccination, SARS-CoV-2 infection, safety

## Abstract

In this study, we report the safety of coronavirus disease 2019 (COVID-19) vaccine in patients with myasthenia gravis (MG). Patients who were vaccinated against COVID-19 were included. Demographics, clinical characteristics, medications, and vaccination information were collected. The main observation outcome is the worsening of MG symptoms within 4 weeks following COVID-19 vaccination. A total of 22 patients with MG vaccinated for COVID-19 were included. Ten (45.5%) patients had ocular MG (OMG), and 12 (55.5%) patients had generalized MG (GMG). Six (27.3%) patients were female, and the mean (SD) onset age was 45.4 (11.8) years. Nineteen (86.4%) patients were seropositive for acetylcholine receptors (AChR) antibody. Seven (31.8%) patients underwent thymectomy, and four of them confirmed thymoma pathologically. Twenty-one patients were administrated with inactivated vaccines, and the remaining one was administrated with recombinant subunit vaccine. Twenty (90.9%) patients did not present MG symptom worsening within 4 weeks of COVID-19 vaccination, and two (9.1%) patients reported slight symptom worsening but resolved quickly within a few days. Our findings suggest inactivated COVID-19 vaccines might be safe in MG patients with Myasthenia Gravis Foundation of America (MGFA) classification I to II, supporting the recommendation to promote vaccination for MG patients during the still expanding COVID-19 pandemic.

## 1. Introduction

Since its outbreak in late December 2019, the pandemic of coronavirus disease 2019 (COVID-19), caused by severe acute respiratory syndrome coronavirus 2 (SARS-CoV-2), had a significant impact on global public health and social order [1]. At the time of 30 July 2021, there were more than 197 million COVID-19 cases causing over 4,207,474 deaths worldwide. The direct and indirect costs of controlling the spread of the disease were immeasurable. Even though enormous effort has been devoted, evidence-based specific therapy for COVID-19 is still scarce [2,3,4,5]. Thus, in the absence of effective treatment, vaccination remains the most effective measure to prevent COVID-19 infection. 

The development of vaccines against SARS-CoV-2 is proceeding at an unprecedented speed. As of 30 June 2021, there were a total of 272 candidate vaccines that were being developed worldwide. Of these, 88 vaccines were in clinical evaluation, and 16 were approved for marketing or emergency use. Results from clinical trials demonstrated the efficacy and safety of COVID-19 vaccines. It was reported that one of the mRNA vaccines was almost 95% effective in preventing COVID-19 in persons 16 years of age or older [6,7,8]. At present, eight vaccines have been included in the WHO emergency use list, and seven vaccines were licensed and are currently in use in China, including five inactivated vaccines, an adenovirus-vector vaccine, and a recombinant subunit vaccine. Of these vaccines, China National Biotec Group (CNBG) SARS-CoV-2 inactivated vaccine, also known as the Sinopharm COVID-19 vaccine and CoronaVac vaccine (Sinovac Biotech Ltd., Beijing, China), have been adopted for mass vaccination within mainland China, and 1049 million doses of inactivated SARS-CoV-2 vaccines have been used as of 21 June 2021. Animal experiments and phase 1 and 2 clinical trials have consistently demonstrated that inactivated SARS-CoV-2 vaccines had notable immunogenicity and a low rate of adverse reactions [9]. A recent cross-sectional study showed side effects of the Sinopharm COVID-19 vaccination for the first and second doses were mild and predictable, and there were no hospitalization cases [10].

The Delta variant (the B.1.617.2 variant), emerging as a virulent strain of COVID-19, caused the number of infections to surge again worldwide, even in some countries with the highest vaccine coverage. The Delta variant strain has stronger transmission capacity and markedly increased virulence, so understanding the efficacy of COVID-19 vaccines against the Delta strain has become the top priority. A real-world study from China showed that the two-dose dosing scheme of the inactivated vaccines was effective against the Delta variant infection, with the estimated efficacy exceeding the World Health Organization’s minimum threshold of 50% [11]. Additionally, a study in Chile spanning from 2 February through 1 May 2021 demonstrated that the effectiveness of CoronaVac vaccine was estimated to be 65.9% for the prevention of COVID-19, 87.5% for the prevention of hospitalization, 90.3% for the prevention of intensive care unit (ICU) admission, and 86.3% for the prevention of COVID-19-related deaths [12].

However, the rapid development of COVID-19 vaccines has raised the question about vaccine safety and effectiveness in immunocompromised patients, such as myasthenia gravis (MG) patients or patients on immunosuppressive medications because such patients were excluded from these vaccine trials. Whether vaccination increases the risk of symptoms worsening or immunosuppressive medications decrease serological responses to vaccines remains unknown. Additionally, the COVID-19 vaccine repertoire is becoming increasingly complex. The novelty of mRNA, DNA, and viral vector COVID-19 vaccines makes it difficult to predict their safety in MG. Currently, there are no reliable data to inform decisions regarding the safety of the COVID-19 vaccine in patients with MG, and neuromuscular experts fail to offer guidance on vaccination in MG patients during the COVID-19 pandemic. 

In this paper, we reviewed the vaccinated MG patients in our database and reported their disease information and clinical course post-vaccination, aiming to evaluate the safety of the COVID-19 vaccine in MG patients.

## 2. Methods

### 2.1. Study Design and Population

This was a retrospective, single-center case series. We reviewed the data of patients in our prospective MG database between 1 July 2020 and 31 May 2021. Patients who were vaccinated against COVID-19 were included, and the physicians in charge confirmed again all case reports from information by phone calls or WeChat (an instant chat tool). 

The diagnosis of MG was considered on the basis of having fluctuating skeletal muscle weakness and at least one of the following criteria: (1) positive response to neostigmine testing; (2) findings suggestive of MG on repetitive nerve stimulation (RNS) findings (more than a 10% decrease in the amplitude of the compound muscle action potential upon stimulation of the facial, ulnar, axillary, and accessory nerves); (3) seropositive for AChR or muscle-specific tyrosine kinase (MuSK) antibodies.

The main observation outcome is the worsening of MG symptoms within 4 weeks of the COVID-19 vaccination, which is defined as the patient’s subjectively reported exacerbation of MG symptoms or the Myasthenia Gravis Activity of Daily Living (MG-ADL) score increasing by ≥2 points compared with before vaccination.

### 2.2. Data Collection

Demographics, clinical characteristics, medications data, and vaccination information were collected: gender; age; onset age; MG type (OMG, GMG); thymoma defined pathologically in patients undergoing thymectomy and radiologically in patients performing a chest imaging test; immunosuppressive drugs before vaccination; types of COVID-19 vaccine (inactivated vaccine, adenovirus-vectored vaccine, recombinant subunit vaccine); and vaccination status (finished vaccination, unfinished vaccination, the date of administration of first and second vaccine doses). Disease status before vaccination was evaluated for symptoms severity. Symptom stability was defined as complete resolution of symptoms for at least 1 month before vaccination; for patients with unstable symptoms, the Myasthenia Gravis Foundation of America (MGFA) classification was used to evaluate the severity of disease [13]. Briefly, MGFA classification I is defined as weakness isolated to ocular muscles and MGFA classification II as mild weakness involving any other muscles. MGFA classifications III and IV are defined by moderate and severe muscle weakness, respectively. MGFA classification V is defined as myasthenic crisis involving respiratory failure requiring endotracheal intubation or non-invasive positive pressure mechanical ventilation. Descriptive analyses were performed using the statistical software package R (version 4.0.0).

### 2.3. Patient Consent and Protocol Approvals

All patients had signed an informed consent form when their information was first registered in the database and authorized the use of clinical information without identification. When the doctor confirmed the information via phone or WeChat, oral informed consent was obtained. The study was approved by the Ethics Committee of the Tangdu Hospital, Fourth Military Medical University (K202107-17).

## 3. Results

A total of 22 patients with MG receiving the COVID-19 vaccine were included. Ten (45.5%) patients had ocular MG, and 12 (55.5%) patients had generalized MG. Six (27.3%) patients were female, the mean (SD) age was 51.1 (12.1) years, and the mean (SD) onset age was 45.4 (11.8) years. Nineteen (86.4%) patients were seropositive for AChR antibody. Seven (31.8%) patients underwent thymectomy, and in four of them, a thymoma was confirmed pathologically. At the time of COVID-19 vaccination, 18 (81.8%) patients had symptoms stability; the median (IQR) symptoms stability time was 15.0 (6.0–24.0) months. The remaining four patients still had MG symptoms, and of them, three patients (No. 7, No. 10 and No. 21) were MGFA classification I, and one (No. 5) patient was MGFA classification IIB. Seventeen (77.3%) patients were taking oral immunosuppressive drugs, including 12 azathioprine, 2 mycophenolate mofetil, and 3 steroids combined with azathioprine. The median (IQR) drug dose stabilization time was 12.0 (6.0–15.0) months. The baseline demographic and clinical characteristics of included patients are summarized in Table 1. Table 2 shows the clinical characteristics of individual patient.

Twenty-one patients were administrated with inactivated vaccines, and the remaining one patient was administrated with a recombinant subunit vaccine (No. 19). No patient in our cohort was administrated with an adenovirus-vectored vaccine. Fifteen (68.2%) patients finished vaccination (two doses of inactivated vaccine), and seven (31.8%) patients unfinished vaccination (one dose of inactivated and recombinant subunit vaccine).

Twenty (90.9%) patients did not present MG symptoms worsening within 4 weeks of COVID-19 vaccination, and two (9.1%) patients reported slight symptom-worsening. Patient No. 19 reported mild neck muscle weakness on the seventh day after the first dose of the recombinant subunit vaccination. Then she increased the dose of pyridostigmine, and the symptoms resolved within a few days. It is noted that she reduced the pyridostigmine dose at the time of vaccination. Patient No. 20 reported mild neck and limb weakness on the 20th day after the first dose of inactivated vaccination. Two days later, he received the second dose of the inactivated vaccine. He started pyridostigmine treatment 8 days after the symptoms worsened, and the symptoms quickly resolved. On the 40th day following the first dose of inactivated vaccination, his MG-ADL score was 3 points, and his QMG score was 8 points during a face-to-face encounter. The symptoms of these two patients were stable for at least 18 months, and thymoma was confirmed pathologically before vaccination.

## 4. Discussion

In this study, we first described the safety profile of the COVID-19 vaccine in patients with MG. Our findings showed most of the MG patients did not present worsening symptoms following COVID-19 vaccination. Even though two patients presented slight symptom-worsening, it resolved rapidly. All these findings indicated that that COVID-19 vaccine might be safe for MG patients with MGFA classification I to II.

As is well-known, the most common trigger of symptom exacerbation in patients with MG is infection [14]. The global COVID-19 pandemic has exposed many immunosuppressed MG patients at risk of infection. Guilhem Solé et al. observed 34 MG patients with COVID-19 in France, of which 28 (82%) patients recovered from COVID-19, and 5 (15%) died [15]. An observational study conducted by Michala Jakubíková et al. in the Czech Republic included 93 MG patients with COVID-19: 35 (38%) developed severe pneumonia after infection, and 10 (11%) patients died due to COVID-19 [16]. A reported registry study named “CARE-MG” launched on April 9, 2020. Ninety-one MG patients had been recorded within 6 months after the start of the study. Of these, 36 (40%) patients reported MG worsening or crisis requiring rescue therapy (e.g., intravenous immunoglobulin, plasma exchange, or steroids) in the setting of COVID-19. Complete recovery or discharge to home was reported in 39 (43%) patients, whereas 22 (24%) patients died due to COVID-19 [17]. Although there is currently no global large sample study to further describe the condition and prognosis of MG patients infected with SARS-CoV-2, it can be seen from results of above studies that MG patients with COVID-19 are more likely to develop severe pneumonia and myasthenia crisis and have relatively high mortality rate. Therefore, it is particularly important to vaccinate MG patients to protect them from COVID-19.

Theoretically, vaccines may trigger or aggravate autoimmune diseases [18]; therefore, it is reasonable to question the safety of COVID-19 vaccine in patients with MG. Several previous studies have evaluated the safety of various vaccines in autoimmune diseases; however, no strong evidence showed that vaccination was associated with the aggravation of autoimmune diseases [19]. Strijbos et al. observed 47 MG patients with the seropositive AChR antibody who received the influenza vaccine or placebo, and they failed to show that influenza vaccination resulted in symptom exacerbation in these patients [20]. In the study of vaccinations with systemic lupus erythematosus, dermatomyositis/polymyositis, and Sjogren’s syndrome, vaccination did not worsen the autoimmune diseases [21,22,23]. COVID-19 vaccine studies in multiple sclerosis (MS) patients showed that RNA, DNA, protein, and inactivated vaccines are likely safe for MS patients. Despite a few incidences of central demyelination being reported with viral vector vaccines, their benefits likely outweigh their risks if alternatives are unavailable. Live-attenuated vaccines should be avoided whenever possible in DMT-treated patients [24]. In our case series, the symptoms of MG did not worsen after inactivated COVID-19 vaccination. Interestingly, the only patient receiving the recombinant subunit vaccine presented worsening symptoms.

The emergency pandemic situation accelerated the development process of the recombinant subunit vaccine, including a nucleic acid-based (mRNA/DNA) coronavirus vaccine, a protein-based coronavirus vaccine, vectored vaccines against coronavirus, etc. Although the safety profile of recombinant vaccines has been demonstrated in a variety of clinical trials [6,7,8], safety data are still lacking for patients with MG to date. Watad et al. reported original autoimmune diseases exacerbation or new disease onset following recombinant mRNA/DNA SARS-CoV-2 vaccination in 27 subjects, including 2 new-onset MG [25]. The two patients developed myasthenia symptoms after the second dose of vaccination. One of the patients first developed ocular symptoms and then rapidly progressed respiratory muscles. The other patient had mild myasthenia symptoms and resolved quickly with plasma exchange and prednisone treatment. Tagliaferri et al. reported a case of myasthenia gravis crisis induced by the Moderna COVID-19 vaccine. The patient was diagnosed with MG five years ago and has been maintained on pyridostigmine and low-dose prednisone. One week after receiving the second dose of the Moderna COVID-19 vaccine, his symptoms worsened and progressed to myasthenia crisis. After being treated with intravenous immunoglobulin, endotracheal intubation, and mechanical ventilation, the patient’s symptoms improved, and the endotracheal tube was successfully removed. The patient’s deterioration was attributed to the second dose of the vaccine [26]. From the results of our study and COVID-19 vaccine studies on other autoimmune diseases, it can be inferred that inactivated vaccines might be safe in MG patients, and the safety profile of mRNA, DNA, and viral vector COVID-19 vaccines should be further investigated.

Previous studies did not report the association of symptoms exacerbation post-vaccination with disease status pre-vaccination [20,27,28,29,30]. Ideally, MG patients should be vaccinated when their symptoms are completely stable. The clinical trial performed by Tackenberg et al. included patients with stable symptoms for more than 4 months only [31], and the Strijbos et al. study included patients with stable symptoms lasting for more than 3 months [20]. In our case series, there were four patients whose symptoms were unstable at the time of COVID-19 vaccination. Among these four patients, No. 5, No. 7, and No. 21 with MGFA classification I presented fluctuating diplopia, and No. 10 with MGFA classification IIB presented intermittent diplopia and slight speech and swallowing weakness. However, all of them did not show symptoms worsening after vaccination. In contrast, two patients (No. 19 and 20) with symptoms worsening post-vaccination reached minimal manifestations status or better at least 18 months before vaccination. Although this case series is not sufficient to draw firm conclusions, the findings showed that worsening symptoms post-vaccination might not be related to disease status pre-vaccination.

As well as the safety profile, vaccine effectiveness is an important concern in these immunocompromised patients, as medications such as rituximab and methotrexate reduce humoral responses and suppress the production of neutralizing antibodies [32,33]. Most MG patients need immunosuppressive therapy for several years or even life-long, including corticosteroids and non-steroidal immunosuppressive drugs or new biologics against B cells and complement. Studies in MS patients demonstrated that interferon-beta, glatiramer acetate, teriflunomide, fumarates, and natalizumab did not impact vaccine efficacy, while cell-depleting agents (ocrelizumab, rituximab, ofatumumab, alemtuzumab, and cladribine) and sphingosine-1-phosphate modulators likely attenuated vaccine responses [24]. Unfortunately, we did not evaluate the effectiveness of COVID-19 vaccines in MG patients. Whether patients with MG or patients on immunotherapy can build up a normal serum titer of anti-COVID-19 antibodies remains to be determined, but under the epidemic situation, even a lower titer of neutralizing antibodies is better than no vaccination.

Our study has certain limitations. Firstly, only patients whose vaccination information was recorded in our database were included, and patients who did not report their vaccination information were not included in this case series, which may lead to an underestimation of the risk of COVID-19 vaccination. Secondly, the sample size of our study is limited, yet it is the first report on this subject. Lastly, the data on the efficacy of COVID-19 vaccines were not collected.

## 5. Conclusions

In the absence of a randomized, controlled trial of COVID-19 vaccination in patients with MG, observational studies are needed to inform practice. Our findings suggest inactivated COVID-19 vaccines might be safe in MG patients with MGFA classification I to II, supporting the recommendation to promote vaccination for MG patients during the still expanding COVID-19 pandemic.

## Figures and Tables

**Table 1 vaccines-09-01112-t001:** Baseline Characteristics of Patients with Myasthenia Gravis Receiving the COVID-19 Vaccine.

Characteristics	Total (*n* = 22)
Gender, n (%)	
Female	6 (27.3)
Male	16 (72.7)
Age, y, mean (SD)	51.1 (12.1)
Onset age, y, mean (SD)	45.4 (11.8)
AChR antibody, n (%)	
Seropositive	19 (86.4)
Seronegative	3 (13.6)
MG type, *n* (%)	
Ocular MG	10 (45.5)
Generalized MG	12 (54.5)
Thymectomy, *n* (%)	
Yes	7 (31.8)
No	15 (68.2)
Thymoma, *n* (%)	
Yes	4 (18.2)
No	18 (81.8)
Disease status before vaccination, *n* (%)	
Stability	18 (81.8)
MGFA I	3 (13.6)
MGFA II b	1(4.6)
Duration of symptoms stable, m, median (IQR)	15.0 (6.0, 24.0)
Immunosuppression protocols, *n* (%)	
Steroids + IS	3 (13.6)
IS monotherapy	14 (63.7)
No therapy	5 (22.7)
Types of immunosuppressive drugs, *n* (%)	
Azathioprine	15 (88.2)
Mycophenolate mofetil	2 (11.8)
Duration of immunosuppressive therapy, m, median (IQR)	12.0 (6.0, 15.0)
Duration of azathioprine treatment, m, median (IQR)	12.0 (5.0, 13.5)
Dose of azathioprine, mg, mean (SD)	75.0 (37.8)
Pyridostigmine	
Yes	9 (40.9)
No	13 (59.1)
Duration of pyridostigmine treatment, m, median (IQR)	4.5 (2.0, 11.5)
Dose of pyridostigmine, mg, mean (SD)	90.0 (40.0)
Type of COVID-19 vaccine, *n* (%)	
Inactivated vaccine	21 (95.5)
Recombinant vaccine	1 (4.5)
Vaccination status, *n* (%)	
Finished vaccination	15 (68.2)
Unfinished vaccination	7 (31.8)
Worsening of MG symptoms	
Yes	2 (9.1)
No	20 (90.9)

Abbreviations: COVID-19 = coronavirus disease 2019; n = number; y = year; SD = standard deviation; AChR = acetylcholine receptor; MG = myasthenia gravis; MGFA = Myasthenia Gravis Foundation of America; IQR = interquartile range; IS = immunosuppressant; m = month.

**Table 2 vaccines-09-01112-t002:** Clinical features of individual patient.

ID	Gender	Age (y)	Onset Age (y)	AChR Antibody	MG Type	Thymoma	Thymectomy	Disease Status before Vaccination	Duration of Symptoms Stability (m)	IS Therapy	Duration of IS Therapy (m)	Drug Dose (mg/d)	Duration of Pyridostigmine (m)	Pyridostigmine Dose (mg/d)	Type of Vaccines/Manufacturer	Vaccination Status	Worsening of MG
1	Male	45	41	Seronegative	GMG	No	Yes	Stability	6	AZA	12	100			Inactivated/CoronaVac	Unfinished	No
2	Male	43	39	Seropositive	GMG	Yes	Yes	Stability	12	AZA	12	100			Inactivated/CoronaVac	Finished	No
3	Male	52	49	Seropositive	GMG	No	No	Stability	20	AZA	20	100			Inactivated/CoronaVac	Finished	No
4	Male	52	49	Seropositive	GMG	No	No	Stability	12	AZA	12	50	12	90	Inactivated/CoronaVac	Unfinished	No
5	Female	73	63	Seronegative	OMG	No	Yes	MGFA II	NA	No	NA				Inactivated/CoronaVac	Unfinished	No
6	Male	53	49	Seropositive	OMG	No	No	Stability	3	Steroid + AZA	4	P 15 mg AZA 100 mg			Inactivated/CoronaVac	Unfinished	No
7	Male	54	53	Seronegative	OMG	No	No	MGFA I	NA	No	NA		1	180	Inactivated/CoronaVac	Finished	No
8	Male	43	41	Seropositive	OMG	Yes	Yes	Stability	2	Steroid + AZA	2	P 2.5 mg AZA 100 mg	2	90	Inactivated /CoronaVac	Finished	No
9	Male	43	38	Seropositive	OMG	No	No	Stability	48	AZA	6		6	60	Inactivated/CoronaVac	Finished	No
10	Female	55	38	Seropositive	GMG	No	No	MGFA I	NA	AZA	3		3	60	Inactivated/CoronaVac	Finished	No
11	Male	58	53	Seropositive	GMG	No	No	Stability	24	AZA	15				Inactivated/CoronaVac	Finished	No
12	Male	60	56	Seropositive	OMG	No	No	Stability	48	MMF	12	250	12	60	Inactivated/CoronaVac	Finished	No
13	Female	69	66	Seropositive	GMG	No	No	Stability	6	MMF	20				Inactivated/CoronaVac	Finished	No
14	Male	67	65	Seropositive	OMG	No	No	Stability	6	AZA	6				Inactivated/CoronaVac	Finished	No
15	Female	25	23	Seropositive	OMG	No	No	Stability	24	AZA	12		12	120	Inactivated/CoronaVac	Finished	No
16	Male	26	22	Seropositive	GMG	No	No	Stability	12	No	NA				Inactivated/Sinopharm	Finished	No
17	Male	46	35	Seropositive	GMG	No	No	Stability	60	AZA	10		10	60	Inactivated/Sinopharm	Finished	No
18	Female	61	43	Seropositive	GMG	No	No	Stability	120	AZA	36				Inactivated/Sinovac	Unfinished	No
19	Female	52	46	Seropositive	OMG	Yes	Yes	Stability	18	MMF	20	1000	1	60	Recombinant/Zhifei Biological Products	Unfinished	neck weakness (6 days)
20	Male	51	46	Seropositive	GMG	Yes	Yes	Stability	23	No	NA				Inactivated/CoronaVac	Unfinished	Diplopia/limb weakness (10 days)
21	Male	56	50	Seropositive	OMG	No	No	MGFA I	NA	Steroid + AZA	1	P 35 mg AZA 100 mg			Inactivated/Sinopharm	Finished	No
22	Male	39	33	Seropositive	GMG	No	Yes	Stability	2	No	NA		2	120	Inactivated/Sinopharm	Finished	No

Abbreviations: y = year; AChR = acetylcholine receptor; MG = myasthenia gravis; OMG = ocular MG; GMG = generalized MG; MGFA = Myasthenia Gravis Foundation of America; m = month; IS = immunosuppressant; AZA = azathioprine; MMF = mycophenolate mofetil; NA = not available; P = prednisone.

## Data Availability

The data that support the findings of this study are available from the corresponding author upon reasonable request.

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
