# Peer review of "COVID-19 Vaccination in Patients with Myasthenia Gravis: A Single-Center Case Series"

_vaccines, 2021, doi:10.3390/vaccines9101112_

Round 1

Reviewer 1 Report

Estimated Authors of the study "COVID-19 vaccination in patient with myasthenia gravis: A single-center case series",

I've read with interest the present paper reporting on a relatively small (but well circumstated) population of patients affected by MG Who had received Sars-CoV-2 vaccine.

Interestingly, the 22 patients were nearly all treated with inactivated vaccine; the Only individual Who had received a different formulation exhibited some worsening of their symptom, but the study design did not allow any further analysis.

In my opinion, the main limits of this study (i.e. the back of any analysis on the potential risk factors for symptom worsening and disease progressione After vaccination) represented its main strengrh, as the Authors made available some early remarks but avoided the potential trap represented by  hasty analyses, that would be otherwise impaired by the reduced Number of samples.

In conclusione, I recommend minor revisions represented by an improved discussion on the Impact of COVID-19 on individuals affected by MG. In fact the present version of this paper does not ectensively discuss whether the Natural infection has or not a worser Impact on MG patients - this Is qquite obviously of significant importance when dealing with the trade off vaccine / do not vaccine. This theme Is not extensively addressed as It would deserve.

Reviewer 2 Report

Major comments: In the present manuscript the authors investigated safety, in particular worsening of myasthenia gravis symptoms, in 22 patients with myasthenia gravis who received COVID-19 vaccination with either an inactivated vaccine (n=21) or an recombinant subunit vaccine (n=1). Two patients experienced worsening of MG symptoms, but the symptoms resolved quickly.

The most important limitation of the study is that no data on effectiveness or humoral immune response after vaccination are provided. Therefore, it cannot be concluded that inactivated vaccines might be the first consideration for these immunosuppressed patients. As we do not know if the patients had an adequate immune response at all.

Furthermore, the authors should clarify for each individual patient which special vaccine was used (Sinopharm or Sinovac, and which special recombinant subunit vaccine was used). Although Sinopharm and Sinovac are WHO approved, there are some doubts concerning their efficacy against the delta variant in European countries and they are not accepted as COVID vaccines there. Therefore, the authors need to give more information in the introduction on those vaccines and discuss the efficacy and safety of the different type of COVID-19 vaccines in regard of use in immunosuppressed patients in the discussion section.

Improvement of English language throughout the article is needed (particularly the abstract and first paragraph of the results)

Minor comments:

Introduction: Line 34: Reference 2 and 3 should be deleted by Beigel et al. Remdesivir for the treatment of COVID-19 - Final report. N Engl J Med 2020.

Results: Clarify the vaccines in Table 2 for the individual patientsDiscussion: The authors should include and discuss the reference: Tagliaferri et al. A case of COVID-19 vaccine causing a myasthenia gravis crisis. Cureus 2021; 13

Discussion: The authors should include and discuss the reference: Tagliaferri et al. A case of COVID-19 vaccine causing a myasthenia gravis crisis. Cureus 2021; 13

Conclusion can only be drawn on safety. Whitout any efficacy data, the study does not support "first consideration" of inactivated vaccines in MG patients. (see also major comments).

Reviewer 3 Report

In this manuscript, the authors investigated the safety of inactivated COVID-19 vaccine in patients with myasthenia gravis. 20 out of 22 patients did not had worsening of MG symptom after vaccination in their hospital. Their findings support the the recommendation to promote vaccination for 22 MG patients with Myasthenia Gravis Foundation of America (MGFA) classification I to II.

To my impression, the manuscript is presented in a well-organized and logical manner. All the results obtained from their studies show reasonable consistency and their conclusions are supported. The study was well conducted, and I appreciate that the authors have given careful thoughts to evaluate the important questions. The manuscript was well written, covering most important details. I would therefore recommend this manuscript for publication in Vaccines.

Round 2

Reviewer 2 Report

In the revised version the authors addressed most of the reviewer's comments. However, they did not improve the first paragraph of the results.

I recommend to perform the following changes there:

Line 119-120: 10 (45.5%) patients had ocular MG, and 12 (55.5%) patients had generalized MG. 6 (27.3%) patients were female and ....

Line 122-123: 7 (31.8%)patients underwent thymectomy , and in 4 of them a thymoma was confirmed pathologically.

Line 131: Table 2 shows the clinical characteristics of individual patients.
